# ACKR2 in hematopoietic precursors as a checkpoint of neutrophil release and anti-metastatic activity

Matteo Massara [1,2], Ornella Bonavita[1,2], Benedetta Savino[1,2], Nicoletta Caronni[1,2], Valeria Mollica Poeta[1,3], Marina Sironi [1], Elisa Setten[1,2], Camilla Recordati [4], Laura Crisafulli[1,5], Francesca Ficara [1,5], Alberto Mantovani [1,3,6], Massimo Locati [1,2] & Raffaella Bonecchi [1,3]

Atypical chemokine receptors (ACKRs) are regulators of leukocyte traffic, inflammation, and immunity. ACKR2 is a scavenger for most inflammatory CC chemokines and is a negative regulator of inflammation. Here we report that ACKR2 is expressed in hematopoietic precursors and downregulated during myeloid differentiation. Genetic inactivation of ACKR2 results in increased levels of inflammatory chemokine receptors and release from the bone marrow of neutrophils with increased anti-metastatic activity. In a model of NeuT-driven primary mammary carcinogenesis ACKR2 deficiency is associated with increased primary tumor growth and protection against metastasis. ACKR2 deficiency results in neutrophil-mediated protection against metastasis in mice orthotopically transplanted with 4T1 mammary carcinoma and intravenously injected with B16F10 melanoma cell lines. Thus, ACKR2 is a key regulator (checkpoint) of mouse myeloid differentiation and function and its targeting unleashes the anti-metastatic activity of neutrophils in mice.

[1] Humanitas Clinical and Research Center, via Manzoni 56, 20089 Rozzano, MI, Italy. [2] Department of Medical Biotechnologies and Translational Medicine, Università degli Studi di Milano, Via Fratelli Cervi, 93, 20090 Segrate, MI, Italy. [3] Department of Biomedical Sciences, Humanitas University, Via Rita Levi Montalcini, 20090 Pieve Emanuele, MI, Italy. [4] Fondazione Filarete, viale Ortles 22/4, 20139 Milano, Italy. [5] Milan Unit, Istituto di Ricerca Genetica e Biomedica, CNR, via Manzoni 56, 20089 Rozzano, MI, Italy. [6] The William Harvey Research Institute, Queen Mary University of London, Charterhouse Square, London, EC1M 6BQ, UK. Matteo Massara, Ornella Bonavita, Benedetta Savino and Nicoletta Caronni contributed equally to this work. Correspondence and requests for materials should be addressed to R.B. (email: raffaella.bonecchi@humanitasresearch.it)

Chemokines are key components of tumor microenvironment[1,2]. Production of chemokines sustains unresolving chronic inflammation, which promotes carcinogenesis. Moreover, oncogene activation drives chemokine production and leukocyte recruitment in tumors epidemiologically unrelated to inflammatory conditions[3].

The repertoire of chemokines produced in the context of neoplastic tissues shapes the leukocyte infiltrate[1,2]. For instance, CCL2 and CXCL8, and related CC and CXC chemokines, are major determinants of macrophage and neutrophil recruitment, respectively. CCL17 and CCL22 have been associated with recruitment of Th2 and regulatory T (Treg) cells, which tame effective antitumor immunity. Conversely, CXCL10 driven recruitment of CD8 T cells and Th1 cells is a driver of antitumor resistance[1,2,4].

Chemokine and chemokine receptors are important determinants of invasion and metastasis[5]. For instance, the chemokine repertoire expressed by tumor cells impacts on secondary seeding at distant sites such as the brain or lymph nodes[6]. Moreover inflammatory chemokines precondition the metastatic niche and drive tumor cell seeding at sites of secondary localization[7].

Canonical chemokine receptors are seven transmembrane G protein-coupled receptors. Mature leukocyte subsets express distinct repertoires of signaling chemokine receptors[8]. Moreover, the chemokine system includes a limited set (four) of atypical chemokine receptors (ACKRs). ACKRs bind a broad panel of inflammatory (ACKR1 and ACKR2) or homeostatic (ACKR3 and ACKR4) chemokines and regulate ligand availability by acting as decoys, scavengers, transporters, or depots[9−11].

ACKR2 (previously known as D6) acts as a decoy and scavenger for most inflammatory CC chemokines and is expressed by lymphatic endothelial cells, trophoblasts in the placenta, and some leukocytes such as innate B cells and alveolar macrophages[11]. In line with its scavenger function, ACKR2 is a negative regulator of inflammation at different anatomical sites[11]. Inflammation is a key component of the tumor microenvironment[12]. Accordingly, genetic inactivation of ACKR2 unleashes tumor-promoting inflammation in the skin and gastrointestinal (GI) tract[13−15]. Moreover, downregulation of ACKR2 in transformed cells has been associated with tumor progression and oncogene activation in Kaposi sarcoma[16−18].

The present study stems from an unexpected observation. In line with previous carcinogenesis results[13,14], in an oncogene-driven mammary carcinoma model we found that ACKR2-deficient mice show enhanced tumor growth at the primary tumor site. In contrast, ACKR2 gene-targeted mice are protected against metastasis. This unexpected finding, extended to transplanted tumors, prompted a dissection of underlying mechanisms. We found that ACKR2 is expressed in hematopoietic progenitor cells (HPCs) and that it serves as a key regulator (checkpoint) of myeloid differentiation and function. Targeting ACKR2 unleashes the anti-metastatic potential of neutrophils.

## Results

*Ackr2*[−/−] **mice are protected against tumor metastasis**. In order to extend previous studies on ACKR2 in carcinogenesis, we crossed Balb/c *Ackr2*[−/−] mice with Balb/c *NeuT* mice, which overexpress the rat *Her2*/neu oncogene under the mouse mammary tumor virus promoter and spontaneously develop mammary carcinomas closely recapitulating human breast carcinogenesis[19]. We followed primary tumor development measuring time of appearance and volume, and found that in *NeuT/Ackr2*[−/−] mice tumor masses in mammary glands developed earlier (Supplementary Fig. 1a) and reached higher volumes as compared to *NeuT/Ackr2*[+/+] mice (Fig. 1a). This result is in accordance with previous reports showing that ACKR2 genetic deficiency results in increased growth of primary tumors[13,14]. Unexpectedly, lung analysis revealed less metastatic lesions in *NeuT/Ackr2*[−/−] mice as compared to *NeuT/Ackr2*[+/+] mice (Fig. 1b, c).

In an effort to strengthen and extend these findings, the transplanted tumor lines 4T1 (mammary carcinoma) and B16F10 (melanoma) were used. 4T1 tumor cells were transplanted orthotopically, whereas B16F10 melanoma cells were injected intravenously (i.v.) in a classic "artificial" hematogenous metastasis model (see below).

When 4T1 cells were injected into the mammary glands of wild-type (WT) and *Ackr2*[−/−] mice, no difference in primary tumor growth was detected (Fig. 1d), but again the number of spontaneous lung metastasis was significantly lower in *Ackr2*[−/−] mice (Fig. 1e, f). Since 4T1 cells expressed little or no *Ackr2*, before and after in vivo growth (Supplementary Fig. 1B), these results suggested that the regulatory function of ACKR2 on metastasis is not cancer cell-intrinsic.

In order to understand which cells protect mice from metastasis, bone marrow (BM) chimeric mice were orthotopically injected with 4T1 cells. Mice were protected from metastasis only when *Ackr2* was genetically inactivated in the hematopoietic compartment (Supplementary Fig. 1c), demonstrating that protection phenotype was due to hematopoietic expression of ACKR2.

**Increased myeloid cell mobilization in *Ackr2*[−/−] mice**. The two breast cancer models used in our experiments are known to induce expansion and mobilization of myeloid cells, which then promote tumor growth[20,21]. Interestingly, when animals were challenged with the 4T1 sibling cell line 66cl4, which in contrast is unable to induce myeloid cell expansion and is less metastatic[22,23], we did not find any difference in the number of metastatic lesions between WT and *Ackr2*[−/−] mice (Fig. 1f). We therefore focused on effects of ACKR2 genetic inactivation on the myeloid compartment of tumor-bearing mice as a potential mechanism of protection from metastasis.

*NeuT/Ackr2*[+/+] mice, at 25 weeks of age, have an increased number of circulating Ly6C[high] monocytes and Ly6G[+] neutrophils (Fig. 2a, b, respectively; gating strategy in Supplementary Fig. 2a) compared with Balb/c WT mice[20,21]. As we previously reported[24], *Ackr2*[−/−] mice have increased number of circulating Ly6C[high] monocytes (Fig. 2a), while no significant difference was found in the number of neutrophils (Fig. 2b) compared to WT mice. At 25 weeks of age, *NeuT/Ackr2*[−/−] mice presented a further significant increase of blood monocytes and neutrophils (Fig. 2a, b) compared to *Ackr2*[−/−] and *NeuT/Ackr2*[+/+], indicating that the lack of ACKR2 was a further driver of tumor-induced myelopoiesis and/or BM release of myeloid cells.

Increased number of inflammatory monocytes and neutrophils, but not alveolar or interstitial macrophages, were also detected in the lungs of *NeuT/Ackr2*[−/−] as compared to *NeuT/Ackr2*[+/+] mice (Fig. 2c and Supplementary Fig. 2b for the gating strategy), while in basal conditions no difference was found between leukocyte infiltrate in the lung of WT and *Ackr2*[−/−] mice (Supplementary Fig. 2c). A higher number of Ly6G[+] neutrophils in the parenchyma of *NeuT/Ackr2*[−/−] lungs compared to *NeuT/Ackr2*[+/+] mice was also found by immunohistochemistry, confirming the flow cytometry data (Fig. 2d, e). Similar results were obtained analyzing blood and lungs of WT and *Ackr2*[−/−] mice orthotopically injected with 4T1 cells (Supplementary Fig. 2d,e, respectively). In these mice, analysis of myeloid cells in the BM 14 days after tumor injection showed a reduced number of monocytes and neutrophils in *Ackr2*[−/−] compared to

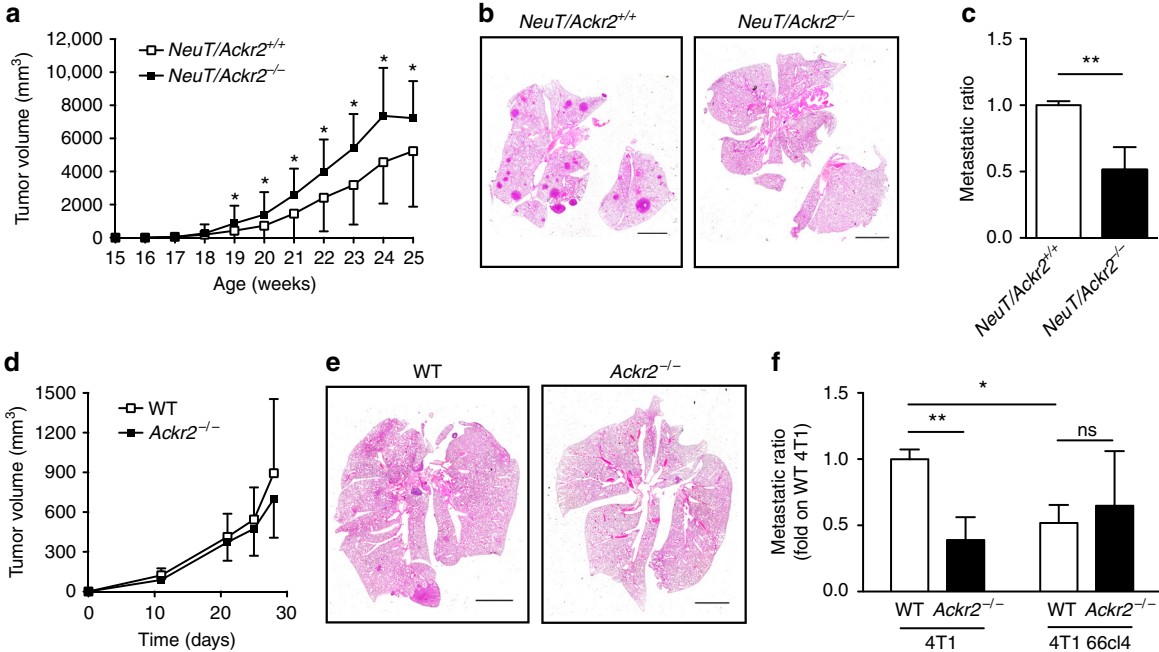

**Fig. 1** $Ackr2^{-/-}$ mice are protected from lung metastasis. **a** $NeuT/Ackr2^{+/+}$ (white squares) and $NeuT/Ackr2^{-/-}$ (black squares) mice were evaluated for tumor growth calculated as described in the Materials and Methods section ($n = 42$ for $NeuT/Ackr2^{+/+}$ and 23 for $NeuT/Ackr2^{-/-}$ mice). **b** Representative images of hematoxylin and eosin staining of $NeuT/Ackr2^{+/+}$ and $NeuT/Ackr2^{-/-}$ lungs at 25 weeks of age. Magnification: ×10. Scale bar: 3 mm. **c** Metastatic ratio of $NeuT/Ackr2^{+/+}$ (white column) and $NeuT/Ackr2^{-/-}$ (black column) mice, calculated as described in the Materials and Methods section ($n = 26$ and $NeuT/Ackr2^{+/+}$ and 16 for $NeuT/Ackr2^{-/-}$ mice, respectively). **d** Tumor volume in WT (white symbols) and $Ackr2^{-/-}$ (black symbols) mice injected orthotopically with 4T1 cells ($n = 14$ for WT and 13 for $Ackr2^{-/-}$ mice). **e** Representative images of hematoxylin and eosin staining of WT and $Ackr2^{-/-}$ lungs at day 28 after 4T1 cell injection. Magnification: ×10. Scale bar: 3 mm. **f** Metastatic ratio of WT (white columns) and $Ackr2^{-/-}$ (black columns) mice at day 28 after orthotopic injection of 4T1 or 4T1 66cl4 cells ($n = 14$ for WT and 13 for $Ackr2^{-/-}$ mice for 4T1, 4 for both WT and $Ackr2^{-/-}$ mice for 4T1 66cl4). Data are represented as mean (SD). $p$ value was generated using the unpaired $t$-test. $^*p < 0.05$, $^{**}p < 0.01$, ns = not statistically different

WT mice (Fig. 2f). These results indicate that in tumor conditions, $Ackr2^{-/-}$ mice show enhanced release of myeloid cells from BM, which then accumulate in the blood and lungs.

**Increased chemokine-induced mobilization in $Ackr2^{-/-}$ mice**. To investigate the role of ACKR2 in myeloid cells egress from the BM, we performed leukocyte mobilization experiments. As previously reported and as shown in Fig. 3a, under homeostatic conditions, ACKR2-deficient mice have increased frequency and absolute number of circulating Ly6C$^{high}$ monocytes compared to WT and a concomitant decrease in the frequency of the same cells in the BM (Fig. 3c), whereas they showed similar number of circulating and BM neutrophils[24] (Fig. 3b, d). After injection of CCL3L1, an ACKR2 ligand known to induce rapid mobilization of both neutrophils and monocytes[25], $Ackr2^{-/-}$ mice showed a significant higher number of circulating monocytes and neutrophils compared to WT littermates (Fig. 3a, b, respectively). Concomitantly, the decrease in monocytes and neutrophils in the BM caused by CCL3L1 injection was more pronounced in $Ackr2^{-/-}$ animals (Fig. 3c, d, respectively).

BM chimera experiments showed that both WT and $Ackr2^{-/-}$ hosts when transplanted with $Ackr2^{-/-}$, but not WT, hematopoietic cells had an increased number of circulating monocytes and neutrophils (Fig. 3e, f; Supplementary Fig. 3a,b) and higher number of monocytes and neutrophils infiltrating the lung (Supplementary Fig. 3c,d). These results demonstrated that the increased mobilization of monocytes and neutrophils induced by CCL3L1 injection was caused by the absence of ACKR2 in the hematopoietic compartment.

Finally, in order to understand whether there is a different localization of leukocytes in the BM sinusoids of ACKR2-

deficient mice, we performed in vivo labeling experiments of monocytes with a 2-minute pulse of anti-Ly6C-PE antibody as previously described[26]. In ACKR2-deficient BM sinusoids there was a significant higher percentage of monocytes located in this vascular compartment after chemokine-induced mobilization compared to WT controls (Supplementary Fig. 3e,f).

**Neutrophils protect $Ackr2^{-/-}$ mice from metastasis**. To investigate the relevance of the increased myeloid cell mobilization found in $Ackr2^{-/-}$ mice in the metastatic process, the B16F10 melanoma cell line was injected i.v. in a classic "artificial" hematogenous metastasis model. In this experimental setting, $Ackr2^{-/-}$ mice showed increased number of circulating neutrophils while no differences were found in the number of circulating monocytes, and T and B lymphocytes compared to WT mice (Supplementary Fig. 4a). Also in this model, there was a significant reduction in the metastatic ratio in the lungs of ACKR2-deficient hosts compared to WT animals (Fig. 4a). In order to understand which cells were responsible for metastasis protection, we performed depletion experiments by using monoclonal antibodies. Monocyte depletion by treatment with an α-CD115 monoclonal antibody significantly decreased the number of metastasis in WT mice, but did not reverse the protection observed in $Ackr2^{-/-}$ mice (Supplementary Fig. 4b). We then performed neutrophil depletion with an α-Ly6G monoclonal antibody. Neutrophil depletion caused a reduction in metastasis in WT mice while an increase in metastatic ratio was observed in $Ackr2^{-/-}$ mice (Fig. 4b). The protective role of neutrophils in $Ackr2^{-/-}$ mice was also demonstrated by performing depletion experiments with the ortotopically transplanted 4T1 tumor line.

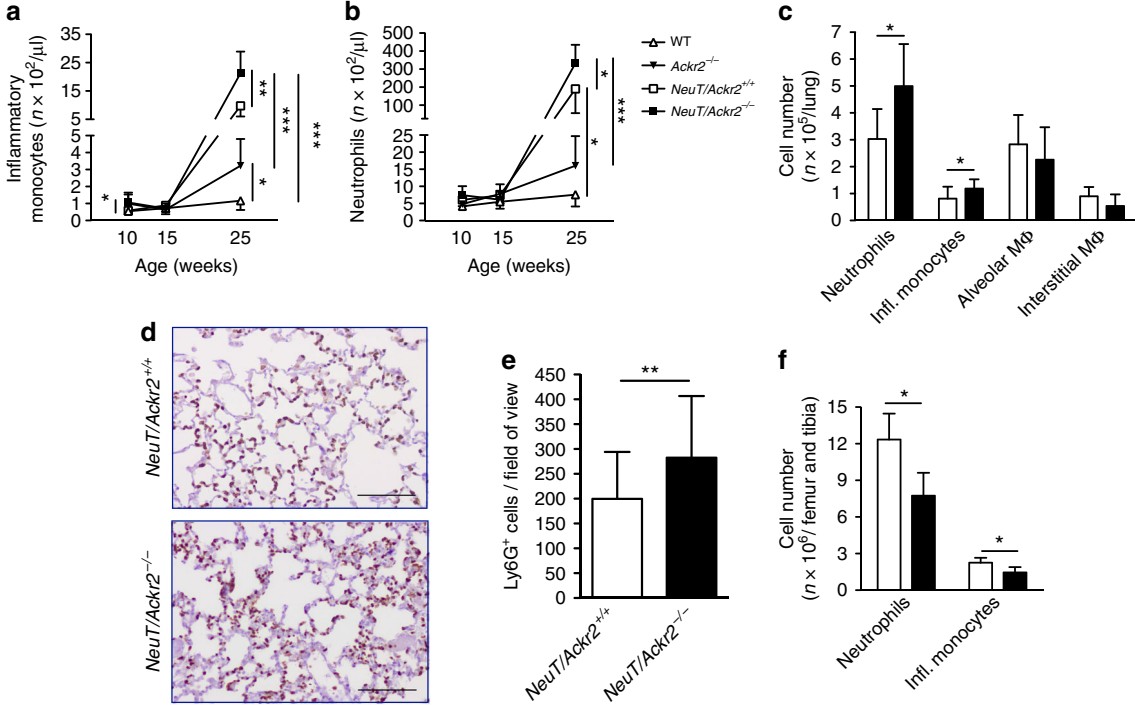

**Fig. 2** Protection from metastasis in $Ackr2^{-/-}$ mice is associated with increased numbers of monocytes and neutrophils in blood and lungs. **a** Absolute number of circulating inflammatory monocytes ($CD45^+/CD11b^+/Ly6C^{hi}$) and **b** neutrophils ($CD45^+/CD11b^+/Ly6G^+$) in $NeuT/Ackr2^{+/+}$ (white squares), and $NeuT/Ackr2^{-/-}$ (black squares), Balb/c WT (white triangles), and $Ackr2^{-/-}$ (black triangles) mice ($n = 9$ for $NeuT/Ackr2^{+/+}$ and 7 for $NeuT/Ackr2^{-/-}$, 3 for both WT and $Ackr2^{-/-}$ at 10 and 15 weeks; 6 for $NeuT/Ackr2^{+/+}$ and 5 for $NeuT/Ackr2^{-/-}$, 5 for both WT and $Ackr2^{-/-}$ at 25 weeks). **c** Absolute number of neutrophils, inflammatory monocytes, alveolar ($CD11b^{low}/F4/80^+/Ly6C^{int}/CD11c^+/Ly6G^-$) and interstitial macrophages ($CD11b^+/F4/80^{int}/Ly6C^-/CD11c^-/Ly6G^-$) in the lungs of $NeuT/Ackr2^{+/+}$ (white columns) and $NeuT/Ackr2^{-/-}$ (black columns) mice at 15 weeks of age ($n = 12$ for $NeuT/Ackr2^{+/+}$ and 6 for $NeuT/Ackr2^{-/-}$ mice). **d** Representative immunohistochemical images of Ly6G staining in $NeuT/Ackr2^{+/+}$ and $NeuT/Ackr2^{-/-}$ lungs at 25 weeks of age. Magnification: ×20. Scale bar: 100 μm. **e** Quantification of Ly6G immunohistochemical images as number of DAB-positive cells on field of view ($n = 9$ for $NeuT/Ackr2^{+/+}$ and 8 for $NeuT/Ackr2^{-/-}$ mice, respectively). **f** Absolute number of neutrophils and inflammatory monocytes in the BM of WT (white columns) and $Ackr2^{-/-}$ (black columns) mice on day 14 after orthotopic injection of 4T1 cells ($n = 4$ for both WT and $Ackr2^{-/-}$ mice). Data are represented as mean (SD). $p$ value was generated using the unpaired $t$-test. $*p < 0.05$, $**p < 0.01$, $***p < 0.001$

Also in this model, neutrophil depletion reduced the metastatic ratio in WT mice, while it increased metastasis in $Ackr2^{-/-}$ hosts (Fig. 4c). Finally, since ACKR2 was reported to be expressed by B lymphocytes[27], we performed B lymphocyte depletion using an α-CD20 monoclonal antibody with no rescue of the metastasis phenotype associated with ACKR2 deficiency (Supplementary Fig. 4c).

The role of $Ackr2^{-/-}$ neutrophils in protection against metastasis was further investigated by adoptive cell transfer experiment. Transfer of $Ackr2^{-/-}$, but not WT, neutrophils into WT tumor-bearing mice significantly reduced the metastatic ratio (Fig. 4d) to values comparable to those observed in $Ackr2^{-/-}$ tumor-bearing mice.

**Increased tumor-killing activity of $Ackr2^{-/-}$ neutrophils**. As depletion and adoptive transfer experiments clearly pointed to neutrophils as the key elements responsible for protection against metastasis observed in the absence of ACKR2, we evaluated their reactive oxygen species (ROS) production, one of the main mechanism of tumor cell-killing. Neutrophils from B16F10-bearing $Ackr2^{-/-}$ animals produced significantly higher amounts of ROS compared to WT mice (Fig. 5a). Similar results were obtained analyzing neutrophils in 4T1-bearing WT and $Ackr2^{-/-}$ mice (Supplementary Fig. 5a).

$Ackr2^{-/-}$ neutrophils also showed a significant increase in transcript levels of the chemokine receptors $Ccr1$, $Ccr2$, and $Ccr5$, but not $Cxcr4$ (Fig. 5b), while expression levels of other genes associated with neutrophils activation, including $Tnf\text{-}\alpha$, $Alox5$,

$Vegf\text{-}a$, and $Arg1$, were not different between WT and $Ackr2^{-/-}$ cells (Fig. 5c). Adoptive transfer experiments demonstrated that $Ackr2^{-/-}$ neutrophils display increased recruitment to the lung compared to WT neutrophils with a concomitant reduction of their number in the blood (Fig. 5d, e, respectively).

Chemokines were reported to control not only neutrophil recruitment to metastatic sites but also their potential anti-metastatic functions such as ROS production[28]. We therefore examined whether the increased levels of CC chemokine receptors found on $Ackr2^{-/-}$ neutrophils could result in increased ROS production. Compared to WT cells, $Ackr2^{-/-}$ neutrophils produced more ROS already under resting conditions and even more after stimulation with the CCR2 ligand CCL2 (Fig. 5f).

In order to assess the functional relevance of this observation, circulating neutrophils were isolated from tumor-bearing mice and evaluated for their ability to kill tumor cells in vitro. Neutrophils obtained from $Ackr2^{-/-}$ mice displayed an increased tumor-killing activity (Fig. 5g) compared to cells isolated from WT mice. The killing activity of both WT and $Ackr2^{-/-}$ neutrophils was partially reversed by the ROS inhibitor apocynin. Similar results were obtained with neutrophils isolated from resting WT and $Ackr2^{-/-}$ mice (Supplementary Fig. 5b).

Collectively, these results suggest that neutrophil activation during tumor progression is constrained by ACKR2, which impinges on their expression of CC chemokine receptors and inhibits their migration to the lung and their ability to generate ROS, key mediators of neutrophil antitumoral potential[29,30].

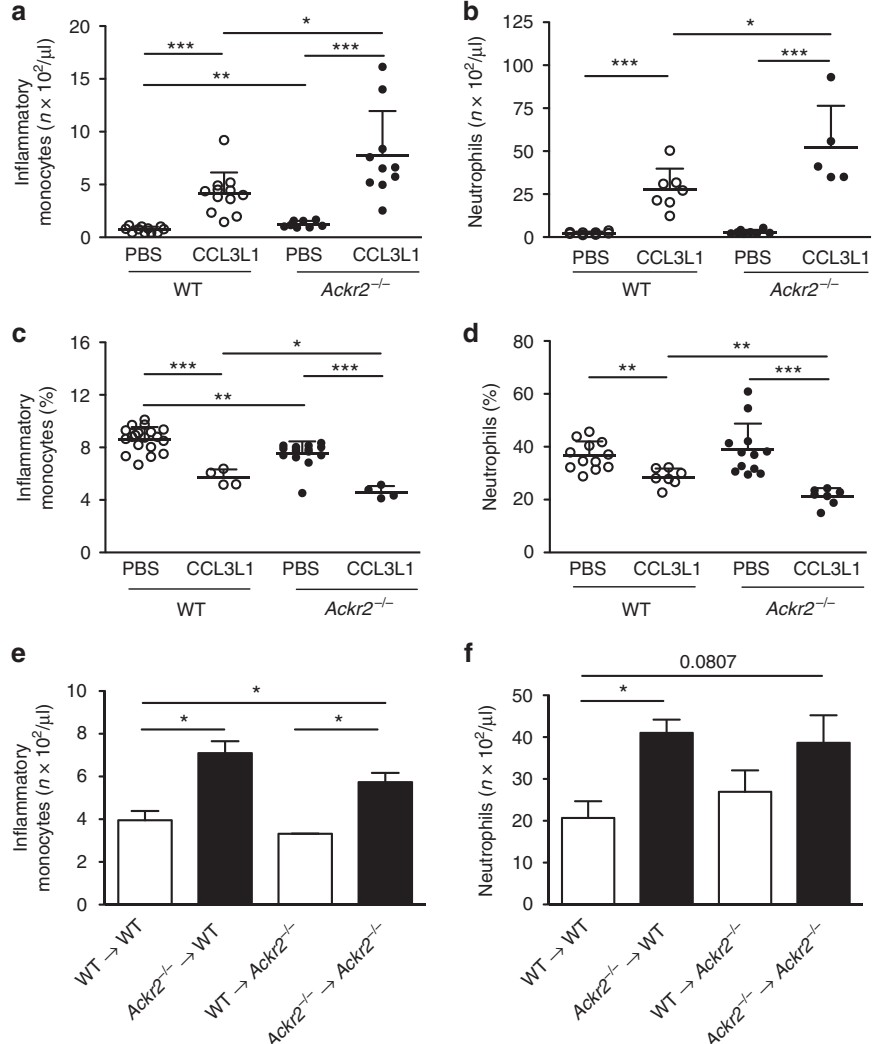

**Fig. 3** Hematopoietic expression of ACKR2 increases monocyte and neutrophil mobilization. Absolute number of inflammatory monocytes (**a**) and neutrophils (**b**) in the blood of C57BL/6J WT or *Ackr2*−/− mice 1 h after i.p. injection of CCL3L1 or vehicle (*n* = 11 for WT/PBS and 8 for *Ackr2*−/−/PBS, 12 for WT/CCL3L1, and 10 for *Ackr2*−/−/CCL3L1 for monocytes; *n* = 8 for both WT/PBS and *Ackr2*−/−/PBS, 7 for WT/CCL3L1, and 5 for *Ackr2*−/−/CCL3L1 for neutrophils). Percentage of monocytes (**c**) and neutrophils (**d**) in the BM of the indicated mice 1 h after CCL3L1 or vehicle i.p. injection (*n* = 19 for WT/PBS and 15 for *Ackr2*−/−/PBS, 4 for both WT/CCL3L1, and *Ackr2*−/−/CCL3L1 for monocytes; *n* = 12 for both WT/PBS and *Ackr2*−/−/PBS, 7 for both WT/ CCL3L1, and *Ackr2*−/−/CCL3L1 for neutrophils). Absolute number of inflammatory monocytes (**e**) and neutrophils (**f**) in WT and *Ackr2*−/− mice reconstituted with either WT (white columns) or *Ackr2*−/− BM (black columns) after i.p. injection of CCL3L1 (*n* = 3 for both WT and *Ackr2*−/− mice). Data are represented as mean (SD). *p* value was generated using the unpaired *t*-test. \**p* < 0.05, \*\**p* < 0.01, \*\*\**p* < 0.001

***Ackr2* is expressed by HPCs and controls myelopoiesis**. Results with BM chimeras demonstrated that ACKR2 expression by hematopoietic cells was responsible for the more pronounced neutrophil mobilization (Fig. 3e, f) and for protection against metastasis found in *Ackr2*−/− mice (Supplementary Fig. 1c). As *Ackr2* mRNA expression was very low in neutrophils (Fig. 6b), we traced its expression on sorted myeloid lineage hematopoietic precursors (gating strategy Supplementary Fig. 6a). *Ackr2* transcript level was highest in Lin^neg/Sca-1+/cKit+ (LSK) cells, and was then downregulated in the more mature common myeloid progenitors (CMPs; Lin^neg/Sca-1−/cKit+/CD34+/ FcγRII/III^int) and granulocytes-monocytes progenitors (GMPs; Lin^neg/Sca-1−/cKit+/CD34+/FcγRII/III^high) (Fig. 6a, b). Thus, the most immature progenitors have the highest level of *Ackr2* expression, which then declines during the myeloid differentiation, showing an opposite behavior as compared to CCR2, whose expression is upregulated during maturation[31] (Fig. 6c).

When we performed expression analysis of other chemokine receptors, we found that, similarly to neutrophils, LSK, CMPs, and GMPs isolated from *Ackr2*−/− mice expressed higher levels of *CCR1*, *CCR2*, and *CCR5* compared to WT mice (Fig. 6c, Supplementary Fig. 6b,c). Fluorescence-activated cell sorting (FACS) analysis confirmed an increased expression of *CCR2* in *Ackr2*−/− LSK, CMP, and GMP cells, and revealed that this was restricted to the myeloid lineage, as no detectable levels of CCR2 were observed in both WT and *Ackr2*−/− megakaryocyte- erythroid progenitors (MEPs; Lin^neg/Sca-1−/cKit+/ CD34−/FcγRII/III−) (Fig. 6d). On the other hand, we did not find differences in CXCR4 mRNA and protein expression in WT and *Ackr2*−/− HPCs (Fig. 6e, f, respectively). These results indicate that ACKR2 is expressed by HPCs and controls the expression of inflammatory CC chemokine receptors known to be involved in myeloid cell release from BM.

WT and *Ackr2*−/− BM did not differ in the absolute number of LSK and CMPs, while a trend of increase was observed in the

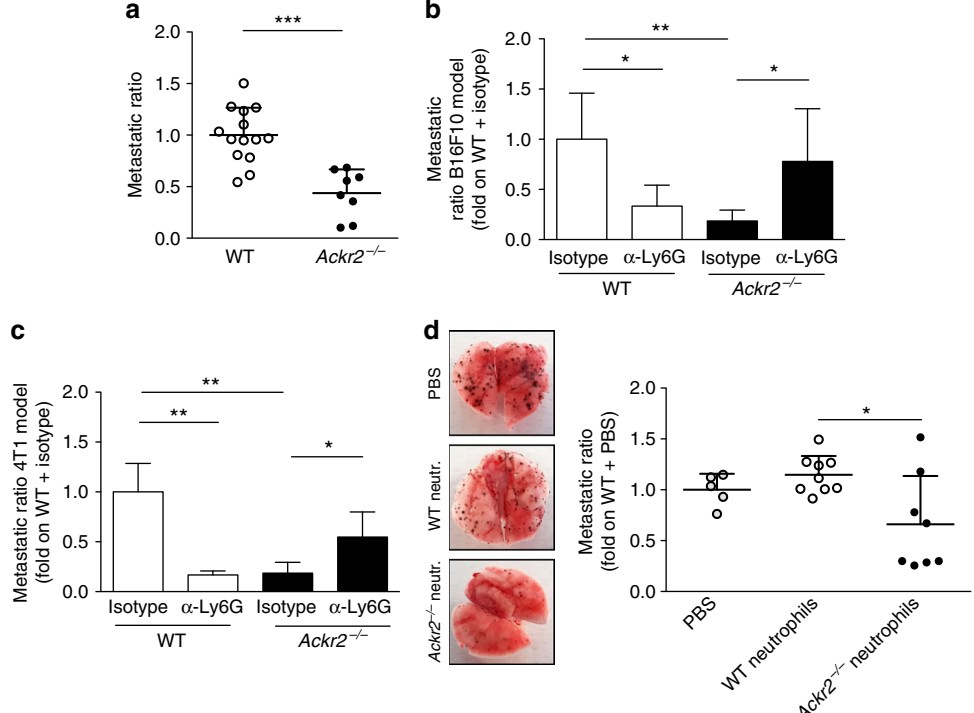

**Fig. 4** *Ackr2*⁻/⁻ neutrophils are responsible for metastasis protection. **a** Metastatic ratio of C57BL/6J WT and *Ackr2*⁻/⁻ mice 10 days after i.v. injection of B16F10 cells ($n = 14$ for WT and 8 for *Ackr2*⁻/⁻, sum of two independent experiments). **b** Metastatic ratio of C57BL/6J WT and *Ackr2*⁻/⁻ mice, treated with isotype IgG or with α-Ly6G, 10 days after i.v. injection of B16F10 cells ($n = 6$ for WT/isotype, and 5 for WT/α-Ly6G, *Ackr2*⁻/⁻/isotype, and *Ackr2*⁻/⁻/α-Ly6G). **c** Metastatic ratio of Balb/c WT and *Ackr2*⁻/⁻ mice, treated with isotype IgG or α-Ly6G antibody, 28 days after orthotopic injection of 4T1 cells ($n = 3$ for WT/isotype, WT/α-Ly6G, and *Ackr2*⁻/⁻/α-Ly6G, and 5 for *Ackr2*⁻/⁻/isotype). **d** Metastatic ratio in WT mice 10 days after i.v. injection of B16F10 cells and adoptive transfer of WT (white dots) or *Ackr2*⁻/⁻ neutrophils (black dots) or PBS (gray dots). Representative images of excised lungs are shown on the left ($n = 5$ for PBS, 9 for WT neutrophils, and 8 for *Ackr2*⁻/⁻ neutrophils). Data are represented as mean (SD). *p* value was generated using the unpaired *t*-test. *$p < 0.05$, **$p < 0.01$, ***$p < 0.001$

number of GMPs in the BM of *Ackr2*⁻/⁻ compared to WT mice (Supplementary Fig. 6d). In addition, no difference in the proliferation of hematopoietic progenitors (Supplementary Fig. 6e) and in the levels of the hematopoietic growth factor granulocyte colony-stimulating factor (Supplementary Fig. 6f) was found comparing resting and tumor-bearing WT and *Ackr2*⁻/⁻ mice.

However, LSK cells sorted from BM *Ackr2*⁻/⁻ and cultured in vitro had increased expression of the myeloid differentiation markers Ly6G, Ly6C, and CD11b, as compared to their WT counterparts (Fig. 6g–i, respectively). Conversely, induction of ACKR2 high expression levels by transfection in the human promyelocytic cell line HL-60, which express low levels of *ACKR2* (Supplementary Fig. 6g,h), resulted in a significant reduction of *CCR2* and *CD11b* expression, while no change in *CXCR4* levels was detectable (Fig. 6j). The increased maturation of neutrophils was also supported by in vivo experiments with a BrdU analog[32]. Indeed, in the blood of *Ackr2*⁻/⁻ mice there were more mature neutrophils compared to WT mice (Supplementary Fig. 6i).

Collectively, these data indicate that ACKR2 genetic inactivation in early hematopoietic precursors results in an accelerated maturation rate of neutrophils, which are more efficiently mobilized by inflammatory chemokines and efficiently recruited to metastatic lesions, where they perform enhanced anti-metastatic activity. ACKR2 in hematopoietic precursors thus operates as a checkpoint for myeloid cell mobilization and effector functions, and its targeting may pave the way to innovative therapeutic strategies, unleashing myeloid cell-mediated protection against cancer.

## Discussion

Chemokines are essential mediators of cancer-related inflammation. Chemokines and chemokine receptors are downstream of oncogene activation and inactivation of oncosuppressor genes, as illustrated by CXCR4 and the VHL pathway[1,33]. Inflammatory chemokines contribute to shaping the landscape of immunity in cancer by recruiting tumor-promoting myeloid cells, polarized Th2 cells, and Treg cells, and by promoting M2-like skewing of tumor-associated macrophages (TAM)[1].

Consistently with this general picture, inactivation of ACKR2, a decoy and scavenger receptor for inflammatory CC chemokines, was associated with enhanced carcinogenesis in the skin and GI tract[13,14]. Conversely, downregulation of ACKR2 driven by Kras activation is associated with tumor progression in human Kaposi's sarcoma and in a mouse vascular tumor model[16]. Tuning of monocyte recruitment underlies the regulatory function of ACKR2 in tumor progression[15].

In agreement with this set of previous data we found that in a model of primary mammary carcinogenesis driven by *Her2*, an oncogene involved in human breast cancer, ACKR2 deficiency was associated with accelerated appearance and growth of primary lesions. Given the tumor-promoting function of TAM in murine and human breast cancer[7,34,35], it is reasonable to assume that enhanced tumor growth was mediated by macrophages. Unexpectedly, in the same model, we found that ACKR2 deficiency was associated with protection against metastasis, an observation in sharp contrast with the primary tumor phenotype.

Protection against lung metastasis in ACKR2-deficient hosts was also observed in the 4T1 mammary carcinoma line

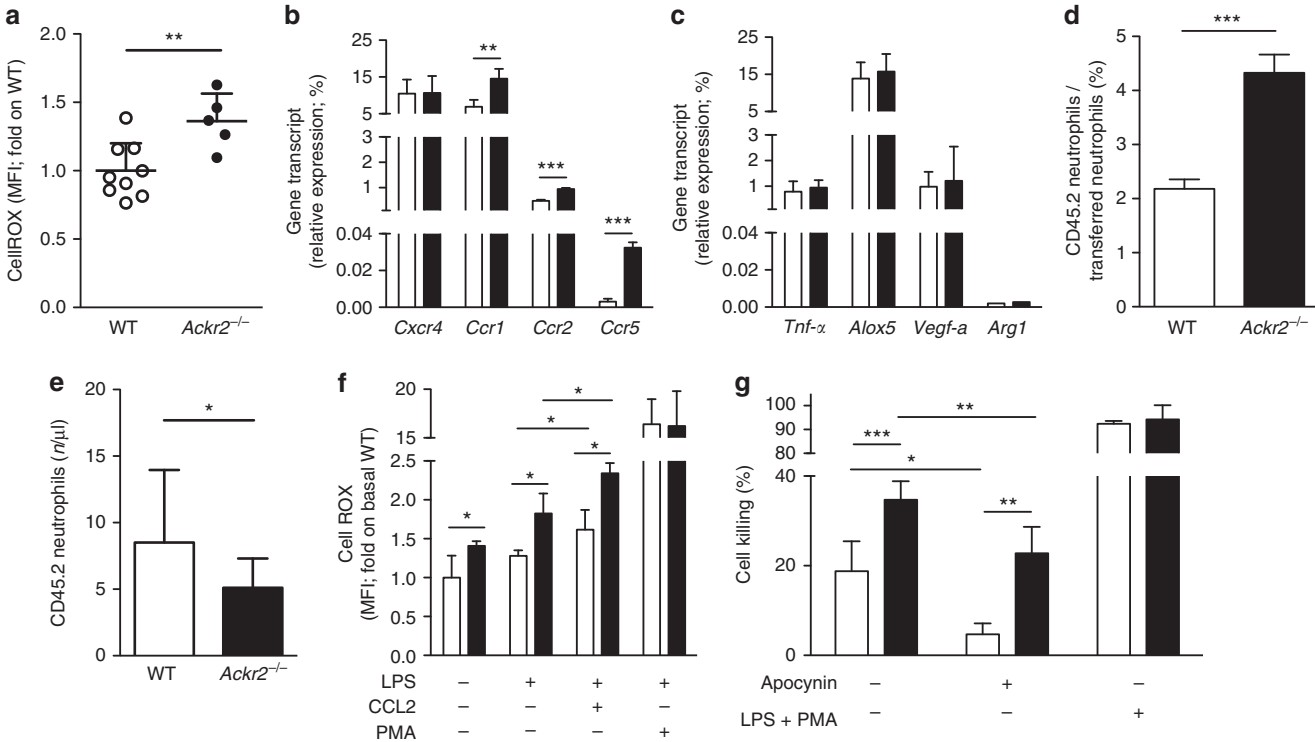

**Fig. 5** $Ackr2^{-/-}$ neutrophils have increased tumor-killing activity and expression of inflammatory CC chemokine receptors. **a** MFI of CellROX emission by WT and $Ackr2^{-/-}$ neutrophils. Data are normalized on MFI of WT neutrophils ($n = 9$ for WT and 5 for $Ackr2^{-/-}$). **b** qPCR analysis of chemokine receptors and **c** activation markers in FACS-sorted neutrophils taken from unchallenged WT (white columns) and $Ackr2^{-/-}$ (black columns) mice ($n = 4$ for both WT and $Ackr2^{-/-}$ mice, two independent experiments). Data are relative to *Gapdh* expression. **d** Percentage of WT and $Ackr2^{-/-}$ CD45.2 neutrophils on total CD45.2 transferred neutrophils in the lungs of WT CD45.1 hosts and **e** absolute number of circulating WT and $Ackr2^{-/-}$ CD45.2 neutrophils after 1 h from adoptive transfer in CD45.1 hosts and i.p. injection of CCL3L1 ($n = 3$ recipients for each group). **f** CellROX MFI in WT (white columns) and $Ackr2^{-/-}$ (black columns) neutrophils preincubated with PBS or LPS (100 ng/ml, 20 min) and stimulated with CCL2 (500 ng/ml, 30 min) or PMA (50 ng/ml, 30 min). CellROX MFI was normalized on basal WT group ($n = 4$, two independent experiments for both WT and $Ackr2^{-/-}$ mice). **g** 4T1-luc cell killing by magnetically sorted circulating neutrophils taken from tumor-bearing WT (white columns) and $Ackr2^{-/-}$ (black columns) mice 21 days after 4T1 injection. Cells were treated with medium or LPS (100 ng/ml) + PMA (50 ng/ml). Where indicated, apocynin (100 µM) was added ($n = 3$, two independent experiments for both WT and $Ackr2^{-/-}$ mice). Data are represented as mean (SD). *p* value was generated using the unpaired *t*-test. *$p < 0.05$, **$p < 0.01$, ***$p < 0.001$

transplanted orthotopically and in the classic B16F10 melanoma cell line injected i.v. The unexpected finding of protection against metastasis prompted a dissection of underlying mechanisms taking advantage of transplanted tumor models.

Several lines of evidence indicate that neutrophils mediate resistance to metastasis in ACKR2-deficient hosts. ACKR2 deficiency was associated with profound alterations in neutrophil phenotype and function. Indeed $Ackr2^{-/-}$ neutrophils had increased ROS production and cell-killing activity, a pattern that identifies mature or activated neutrophils[32]. Moreover, neutrophils lacking ACKR2 had increased expression of the chemokine receptors *Ccr1*, *Ccr2*, and *Ccr5*, a chemokine receptor profile again typical of activated neutrophils. Indeed, neutrophils express different pattern of chemokine receptors depending on their activation state[36,37]. Interferon-γ, the prototypic Th1 cytokine, upregulates the expression of the CC chemokine receptors CCR1 and CCR3[38], and inflammatory stimuli such as lipopolysaccharide (LPS) induce CCR2 expression in neutrophils[39]. The expression of CC chemokine receptors by neutrophils is functionally relevant not only in terms of recruitment to the inflamed site but also in the activation of their effector functions such as respiratory burst, bacterial killing, and anti-metastatic activity[28,36,40]. In line with these data Ackr2-deficient neutrophils showed increased recruitment to the lungs (Supplementary

Fig. 3d) and increased tumor-killing activity (Fig. 5g) compared to WT neutrophils.

The role of neutrophils in protection against metastasis was also demonstrated by depletion experiments. Neutrophil depletion rescued the phenotype of resistance to metastasis observed in ACKR2-deficient hosts. Interestingly, in ACKR2-competent mice neutrophil depletion resulted in reduced metastasis (Fig. 4b, c), in line with several observations of neutrophil-mediated tumor promotion (see below). Finally, adoptive transfer of ACKR2-deficient neutrophils mediated resistance against metastasis (Fig. 4d). Thus, unleashing neutrophil-mediated resistance underlies protection against metastasis observed in ACKR2-deficient mice.

The finding that *Ackr2* expression was vanishingly low in mature neutrophils, raised the possibility of a regulatory function of this molecule upstream in hematopoiesis. We found that *Ackr2* is expressed by HPCs and that it is downregulated during maturation in myeloid progenitors. Interestingly, ACKR2 was cloned in 1997 from the BM[41] but its role and expression in this compartment has not been explored[42]. Here we found that $Ackr2^{-/-}$ HPCs, like $Ackr2^{-/-}$ neutrophils, express extremely high levels of the chemokine receptors CCR1, CCR2, and CCR5 and, when cultured with differentiating cytokines, acquire myeloid differentiation markers faster compared to WT counterparts.

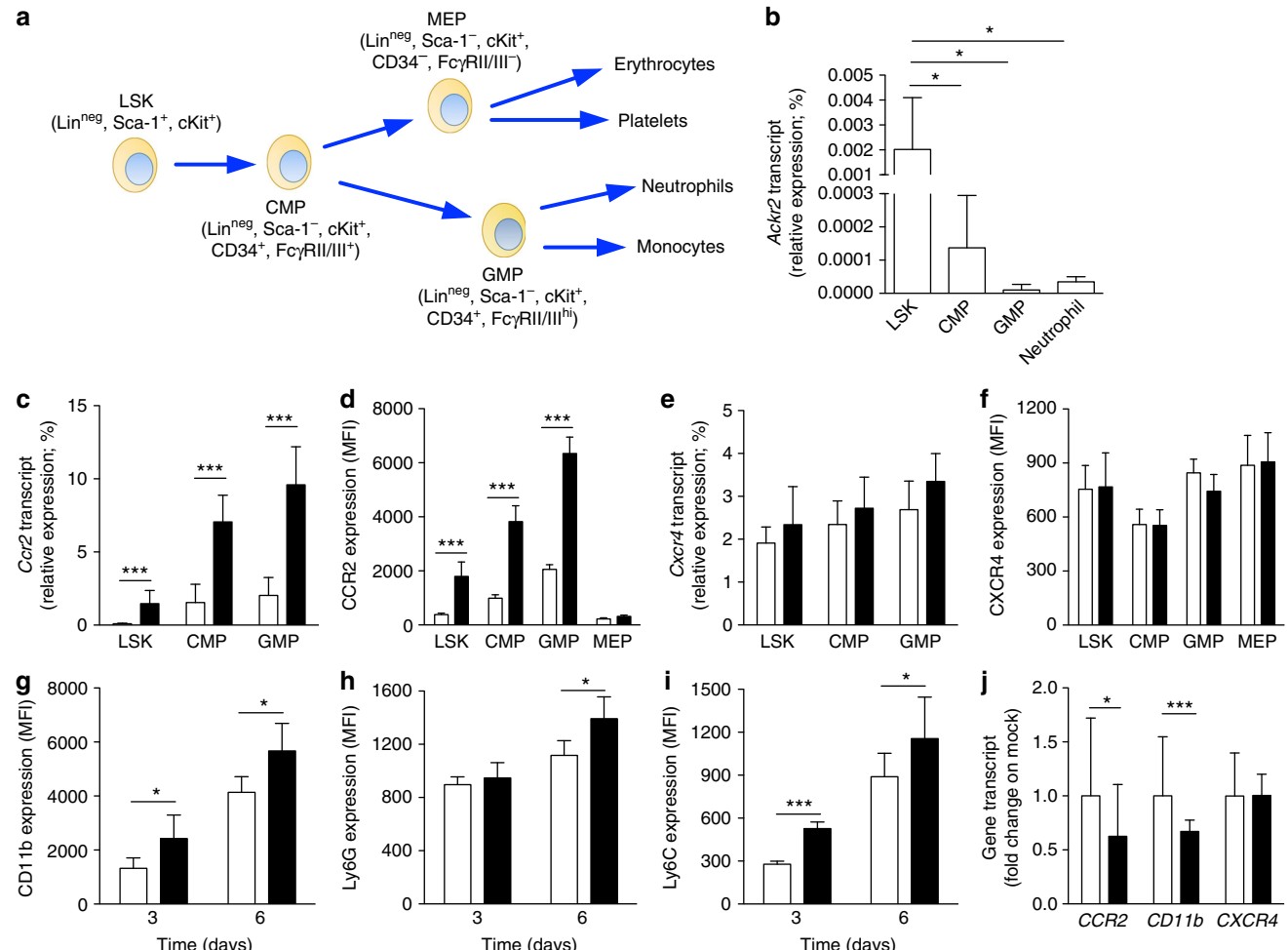

**Fig. 6** *Ackr2* is expressed in HPCs and controls expression of CC chemokine receptors. **a** Schematic representation of murine hematopoietic cell differentiation. **b** qPCR analysis of *Ackr2* expression on sorted HPCs and neutrophils taken from BM of WT mice ($n = 7$). **c** qPCR analysis of *Ccr2* and **e** *Cxcr4* expression on sorted HPCs taken from WT (white columns) and *Ackr2$^{-/-}$* (black columns) mice. All qPCR data are relative to *β-actin* expression ($n = 7$ for both WT and *Ackr2$^{-/-}$* mice). **d** MFI of CCR2 and **f** CXCR4 expression measured by FACS analysis in HPCs taken from WT (white columns) and *Ackr2$^{-/-}$* (black columns) mice ($n = 4$ for both WT and *Ackr2$^{-/-}$* mice). **g** MFI of CD11b, **h** Ly6G, and **i** Ly6C expression on FACS-sorted LSK cells taken from WT (white columns) and *Ackr2$^{-/-}$* (black columns) cultured in vitro as described in Materials and Methods ($n = 5$ for WT and 7 for *Ackr2$^{-/-}$* at 3 days, and 5 for WT and 6 for *Ackr2$^{-/-}$* at 6 days). **j** qPCR analysis of *CCR2*, *CD11b*, and *CXCR4* expression in HL-60 cells transfected with *ACKR2* (black columns) or empty vector (white columns). qPCR data are relative to *GAPDH* expression and normalized on mock transfected cells ($n = 9$ for mock and 13 for *ACKR2*-transfected cells, three independent experiments). Data are represented as mean (SD). *p* value was generated using the unpaired *t*-test. *$p < 0.05$, ***$p < 0.001$

To directly assess the effect of ACKR2 on chemokine receptors in immature hematopoietic elements, the HL-60 cell line was used. Here ACKR2 downregulated the expression of *CCR2* and of the integrin *CD11b*. ACKR2 induces a signal transduction cascade activating a β-arrestin-dependent signaling that optimizes its chemokine scavenging activity[43]. It is likely that in HPCs β-arrestin signaling by ACKR2 negatively controls the expression of CC chemokine receptors by interfering with other signaling pathways or by activating mechanism of negative regulation as previously described for other ACKRs. ACKR3, for example, negatively controls signaling by CXCR4[44] and ACKR4 inhibits the expression of CCR7, CCR9, CXCR5, and CXCR4[45]. Our data are also consistent with previous published data reporting that ACKR2 regulates neutrophil migration toward inflammatory CC chemokines[46].

Neutrophils have emerged as important players in cancer-related inflammation. In general, several lines of evidence indicate that neutrophils are part of the inflammatory network, which promotes tumor progression and metastasis[47–49]. In accordance

with this line of evidence, we observed that neutrophil depletion using an α-Ly6G monoclonal antibody resulted in decreased lung metastasis in the B16F10 and in the 4T1 models (Fig. 4b, c). However, neutrophils can undergo functionally reprogramming in response to tumor- and host-derived signals[48,49] and accordingly exert divergent influence on tumor growth[50], a dichotomy mirrored by prognostic significance in different human cancers[51].

Here we report that *Ackr2* expressed by HPCs is a key setpoint of neutrophil differentiation, mobilization, and function (Supplementary Fig. 7). Targeting hematopoietic ACKR2 may pave the way to innovative therapeutic strategies unleashing myeloid cell-mediated protection against infection and cancer.

## Methods

**Cell lines**. 4T1 and 4T1-66cl4 cells (kindly provided by Dr. Claudia Chiodoni, Department of Experimental Oncology and Molecular Medicine, Istituto Nazionale dei Tumori, Milano, Italy) were grown in Dulbecco's modified Eagle medium (DMEM; Lonza) supplemented with 10% fetal bovine serum (FBS; Sigma), 1% penicillin/streptomycin (Lonza), 1% L-glutamine (Lonza), 1% sodium pyruvate

(Lonza), and 1% Hepes (Lonza). B16F10, kindly provided by Prof. Massimiliano Mazzone (Vesalius Research Center, Leuven, Belgium), were grown in DMEM (Lonza) supplemented with 10% FBS (Sigma), 1% penicillin/streptomycin (Lonza), and 1% L-glutamine (Lonza). 4T1-luc from PerkinElmer were grown in RPMI 1640 (Lonza), 10% FBS (Sigma), 1% penicillin/streptomycin (Lonza), 1% L-glutamine (Lonza), 1% sodium pyruvate (Lonza), and 5.4 g/l glucose (Sigma). HL-60 were purchased from American Type Culture Collection and grown in Iscove's modified Dulbecco's medium (IMDM; Lonza), 20% FBS (Sigma), 1% penicillin/streptomycin (Lonza), 1% L-glutamine (Lonza), and 1% sodium pyruvate (Lonza), and transfected with pEGFP-N1 ACKR2 or mock vector[52] by using the Nucleofector Kits for HL-60 (Lonza) according to the manufacturer's instructions. Green fluorescent protein-positive cells were sorted for mRNA analysis. Cells were tested for mycoplasma and only mycoplasma-free cells were used.

**Animals**. $Ackr2^{-/-}$ mice were maintained on Balb/c and C57BL/6J genetic background. Balb/c WT and $Ackr2^{-/-}$ mice were crossed with NeuT mice (kindly donated by Professor Federica Cavallo, University of Turin, Italy). WT and WT CD45.1 mice were obtained from Charles River Laboratories (Calco, Italy) or were co-housed littermates. All colonies were housed and bred in the SPF animal facility at Humanitas Clinical and Research Center in individually ventilated cages. Mice used for experiments were 8–12 weeks old. Procedures involving animal handling and care were conformed to protocols approved by the Humanitas Clinical and Research Center (Rozzano, Milan, Italy) in compliance with national (4D.L. N.116, G.U., suppl. 40, 18-2-1992) and international law and policies (EEC Council Directive 2010/63/EU, OJ L 276/33, 22-09-2010; and National Institutes of Health Guide for the Care and Use of Laboratory Animals, US National Research Council, 2011). The study was approved by the Italian Ministry of Health (approval no. 88/2013-B, issued on 08 April 2013). All efforts were made to minimize the number of animals used and their suffering. Mice were randomized based on sex, age, and weight. The sample size was chosen on the basis of past experience on tumor models in order to detect differences of at least 20% between the groups. In most in vivo experiments, the investigators were unaware of the genotype of the experimental groups.

**Tissue collection**. Blood was collected from the retro-orbital plexus and by cardiac puncture as described[53]. Briefly, blood was collected in 2KD-EDTA spray-coated tubes (BD Bioscience), washed in FACS buffer (PBS$^{-/-}$, 1% bovine serum albumin, and 0.05 % sodium azide), red blood cells were lysed, washed again, and cells were stained as indicated. Lungs were instilled with PBS for FACS analysis or 4% neutral buffer formalin for histological analysis. For FACS analysis, lungs were minced, digested for 45 min in 1 mg/ml collagenase D (Sigma) in PBS$^{-/-}$, and filtered with 70 μm cell strainer. Red blood cells were lysed and cells stained as indicated. BM was collected from femurs and tibiae. Bones were harvested, cleaned, flushed, and filtered with 70 μm cell strainer. Red blood cells were lysed and cells stained as indicated below.

**Tumor models**. Tumor volume was assessed with caliper using the formula: (length × width × width)/2. Tumor take in NeuT model was determinate by palpation as number of mammary tumors per mouse. For 4T1 and 4T1-66cl4 models $5 \times 10^5$ cells were injected in the mammary fat path of Balb/c mice. For lung metastasis evaluation in NeuT, 4T1, and 4T1-66cl4 models, lungs were instilled and fixed for 24 h with 4% neutral buffered formalin, routinely processed for paraffin embedding, sectioned at 4 μm thickness, and stained with hematoxylin and eosin. Sections were evaluated in a blinded fashion under a light microscope. Lung metastasis in NeuT, 4T1, and 4T1-66cl4 models were classified according to their size into: small (<30 neoplastic cells), medium (30–300 neoplastic cells), and large (>300 neoplastic cells). A total metastatic score was then calculated for each lung as follows: number of small metastases × 1 + number of medium metastases × 3 + number of large metastasis × 5. Representative images were acquired with Slide Scanner VS120 dotSlide (Olympus) and analyzed with ImageJ. The melanoma cell line B16F10 ($2 \times 10^5$ cells) was injected i.v. in C57BL/6 mice and metastases were macroscopically counted as dark nodules on the lung surface. For all the models, metastatic ratio was calculated as ratio of metastasis in $Ackr2^{-/-}$ or depleted mice compared to indicated control mice. For monocyte depletion, mice were treated with 100 μg of α-CD115 antibody (clone AFS98, Bioxcell) the day before $2 \times 10^5$ B16F10 injection and every 2 days for the entire duration of the experiment. For neutrophil depletion, mice were treated with 200 μg of α-Ly6G antibody (clone 1A8, Bioxcell) the day before $2 \times 10^5$ B16F10 injection and with 100 μg every 3 days for the entire duration of the experiment. For B-cell depletion, mice were treated with 250 μg of α-CD20 (clone 5D2, Genentech Inc.) 3 days before $2 \times 10^5$ B16F10 injection. For adoptive transfer experiments, neutrophils were isolated from WT and $Ackr2^{-/-}$ BM using the Mouse Neutrophil Isolation Kit (Miltenyi Biotec) and an autoMACS Pro separator (Miltenyi Biotec). Cell purity was assessed by flow cytometry (CD45, CD11b, and Ly6G) and used only if neutrophils were ≥95% on CD45$^+$ cells. For B16F10 model, recipient WT mice were injected i.v. with $5 \times 10^6$ WT or $Ackr2^{-/-}$ neutrophils every 3 days for the entire duration of the experiment. For adoptive transfer experiment, recipient CD45.1 mice were injected i.v. with $2 \times 10^6$ CD45.2 WT or $Ackr2^{-/-}$ neutrophils 15 min before CCL3L1 (R&D)

injection and after 1 h lung and blood were collected and leukocytes counted by flow cytometry.

**Immunohistochemistry**. Serial 4 μm formalin-fixed and paraffin-embedded lung sections were deparaffinized and underwent heat-induced epitope retrieval with pressure cooker. Endogenous peroxidase activity was blocked by incubating sections in 3% $H_2O_2$ for 15 min. Slides were rinsed and treated with Rodent Block M (Biocare Medical) for 30 min to reduce nonspecific background staining and then incubated for 1 h at room temperature with Ly6G antibody (1:200; clone 1A8; BD Bioscience), Sections were incubated for 30 min with Rat on Mouse HRP-Polymer kit (Biocare Medical). The immunoreaction was visualized with 3,3'-diaminobenzidine (Peroxidase DAB Substrate Kit, Vector Laboratories) substrate and sections were counterstained with Mayer's hematoxylin. Negative immunohistochemical controls for each sample were prepared by replacing the primary antibody with normal serum. Positive control sections were included in each immunolabeling assay. Tissues were dehydrated with ethanol, mounted with Eukitt, and acquired with an Olympus BX61 virtual slide scanning system using Cell^F software (Olympus). In each section 10 independent field of view were acquired. To evaluate the extent of granulocytes infiltration in the lung parenchyma, the percentage of Ly6G-positive area was analyzed with Image-Pro Analyzer 7.0 (Media Cybernetics) software. Representative images were generated using the ImageJ analysis program (http://rsb.info.nih.gov/ij/).

**Flow cytometry analysis**. Flow cytometry analysis was performed as previously described[53]. To exclude dead cells from analysis, cells were stained with Violet dead cell stain kit (Thermo Fisher). Single-cell suspension was stained with antibodies listed in Supplementary Table 1 and related isotype. All antibodies were purchased from BD Bioscience, BioLegend, eBioscience, or AbD Serotec. Flow cytometry data were acquired using a FACSCanto II (BD Bioscience) and LSR Fortessa (BD Bioscience), and data were analyzed with FACS Diva (BD Bioscience) and representative images were generated with FlowJo Software (Tree Star). To analyze ROS production, neutrophils were stained with 5 μM CellROX Deep Red Reagent (Thermo Fisher) for 20 min at 37 °C in RPMI 1% FBS. Staining was blocked on ice, red blood cells were lysed, and neutrophils were analyzed by flow cytometry within 2 h from the staining. Where indicated, mice were injected with 500 μg of Click-it EdU Plus (Thermo Fisher) resuspended in PBS. Femurs were harvested 2 h after injection for hematopoietic progenitor analysis. For neutrophil analysis blood was collected 48 and 72 h after injection. Staining was performed according to the manufacturer's instructions. The absolute number was determined by using TruCount beads (BD Biosciences) according to the manufacturer's instructions. Cell sorting was performed using a FACSAria III (BD Bioscience).

**HPC isolation and culture**. Lineage-negative cells were isolated from WT and $Ackr2^{-/-}$ BM using LS columns and the Mouse Lineage Cell Depletion Kit (Miltenyi Biotec), according to the manufacturer's instructions. Negative fraction was stained with Streptavidin-PB, Sca-1, cKit, CD34, and FcγRII/III antibodies and sorted. LSK were seed ($1 \times 10^3$/well) in rounded-bottom 96-well plate in IMDM medium (Lonza) supplemented with 10% FCS (Sigma), 1% L-glutamine (Lonza), 20 ng/ml stem cell factor (SCF) (Peprotech), 10 ng/ml interleukin (IL)-6 (Peprotech), and 10 ng/ml IL-3 (Peprotech), as previously described[54]. Cells were harvested 3 and 6 days after seeding, stained, and analyzed by flow cytometry.

**Leukocyte mobilization**. Mice were injected intraperitoneally with 3 μg CCL3L1 (R&D) and after 1 h blood was collected and leukocytes counted by flow cytometry. To evaluate the percentage of monocytes in BM sinusoids, mice were injected intravenously with 1 μg Ly6C$^-$PE antibody 2 min before the end of experiment.

**Generation of BM chimeras**. Recipient mice received gentamycin (0.8 mg/ml) in drinking water for 2 weeks starting 10 days before irradiation. WT and $Ackr2^{-/-}$ mice were lethally irradiated with a total dose of 900 cGy. After 2 h, mice were injected in the retro-orbital plexus with $4 \times 10^6$ nucleated BM cells obtained by flushing of the cavity of a freshly dissected femur from WT or $Ackr2^{-/-}$ donors. Experiment were performed 16 weeks after irradiation to allow complete myeloid repopulation.

**In vitro cell-killing assay**. Neutrophils were isolated by magnetic separation as described above from blood of 14 days' 4T1 tumor-bearing mice or from BM of untreated mice and seeded ($1 \times 10^5$/well) in a 96-well plate in which, 4 h before, $5 \times 10^3$ 4T1-luc cells were plated in Optimem (Thermo Fisher) + 0.5% FBS. Cells were incubated overnight in presence of apocynin 100 μM (Sigma) or dimethyl-sulfoxide control. Firefly luciferase activity was detected with luciferase assay system (Promega) and Synergy H2 (Biotek). Cell killing was calculated as percentage of tumor lysis by the following formula: % cell killing = (1 − [luminescence of samples with neutrophils]/[luminescence of samples in medium]) × 100.

**Transcript analysis by quantitative PCR**. Total RNA was extracted from HPCs with miRNA easy Mini kit (Qiagen). Reverse transcription was done using High Capacity cDNA Reverse Transcription Kit (Applied Biosystems). Quantitative PCR

(qPCR) was performed with TaqMan Gene Expression Assays (Thermo Fisher) in a 7900 HT Fast Real-Time PCR System (Applied Biosystems) with probes listed in Supplementary Table 2. Total RNA was extracted from HL-60 cells and neutrophils using the TRIzol reagent (Thermo Fisher). Reverse transcription was done using High Capacity cDNA Reverse Transcription Kit (Applied Biosystems). qPCR was performed with TaqMan Gene Expression Assays in a CFX Connect Real-Time PCR Detection System (BioRad) with probes listed in Supplementary Table 2. Relative mRNA expression was determined by using the $2^{-\Delta Ct}$ method, and normalized to the expression of the housekeeping gene *β-actin* or *Gapdh*.

**Statistical analysis**. Data are represented as mean. In all figures sample variation is shown as SD. *p* value was generated using the unpaired *t*-test after normality test and F test to exclude difference in the variance between groups (GraphPad Prism 5). *$p < 0.05$, **$p < 0.01$, ***$p < 0.001$, ns = not statistically different.

**Data availability**. The authors declare that all the data supporting the findings of this study are available within the article and its supplementary information files and from the corresponding author upon reasonable request.

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

## Acknowledgements

This study was supported by the Italian Association for Cancer Research (AIRC—IG 15438) to R.B. and in part by a grant from the Italian Ministry of Health (GR-2010-2307975) to F.F. and by a grant HEALTH-F4-2011-281608 (TIMER) to A.M.. L.C. is recipient of a fellowship from Fondazione Nicola del Roscio. We acknowledge the Humanitas Flow Cytometry and Imaging Core for the technical assistance during the experiments.

## Author contributions

R.B., M.L., A.M., and F.F. conceived and designed the experiments. M.M., O.B., B.S., N.C., M.S., V.M.P., E.S., C.R., L.C., and F.F. performed the experiments. R.B., M.M., O.B., B.S., N.C., C.R., and M.L. analyzed the data. R.B., M.L., and A.M. wrote the paper.

## Additional information

**Competing interests:** The authors declare no competing financial interests.

