## [Peer Review File · Nature Communications]

Reviewer #1:

(Remarks to the Author):

Massara and co-authors investigate the role of an atypical chemokine receptor (ACKR2) in mediating metastasis of tumors. They propose a role for this receptor in modulating myeloid differentiation and subsequently the anti-metastatic activity of neutrophils. Authors use inactivation or downregulation of ACKR2 and injection in vivo models of recombinant ligands.

Suppl figure 4 summarizes the data. Whereas the paper and the proposed mechanism is of potential interest, more experimental data are needed to convincingly and sufficiently support the presented summary scheme and mechanism.

Specific major comments:

1. Page 6: Authors propose increased mobilization of myeloid cells in ACKR2-deficient mice. Does ACKR2-deficiency affect proliferation of myeloid cells and their precursors? It is important to show the respective experimental data on proliferation (and thus exclude this mechanism).
2. On page 7 and in figure 4B authors report that depletion of myeloid cells reduces metastasis in WT mice but induces metastasis in ko mice. How is this explained? The authors do not make sufficient attempts to experimentally address this. They could for example compare the anti-metastatic potential of CD115+ cells and Ly-6G+ cells from WT and ko animals. How do they differ? It seems that this is a key question in this project?
3. Page 8, figure 5 and suppl Figure 4: Authors investigate ROS production and conclude that this is implicated in antimetastatic activity and tumor cell killing. To make this manuscript conclusive it is essential to perform tumor cell killing assays and to show that ROS is indeed involved in killing. Otherwise the experiments shown only present associations and not a mechanism as proposed in suppl figure 4. In addition, do neutrophils from WT and KO animals display different killing activity?
4. Page 9: Authors conclude that neutrophils are more efficiently recruited to metastatic lesions. Could the authors please address this experimentally for example by measuring the recruitment of adoptively transferred neutrophils from WT and KO. At present the authors do not really address the recruitment process.
5. Figure 3: E/F show by BM transfer that expression of ACKR2 on neutrophils/BM affects cell number in blood. However, for this paper, the number of neutrophils in the lungs would be important (site of metastasis). It is important to repeat figure 3 E/F and measure neutrophils in the lungs.
6. Figure 4: Please show absolute numbers of metastasis (not relative score) in order to be able to compare data to (A) and (C).

7. Suppl figure 1: Authors phenotype lung neutrophils without showing Ly-6G (key marker) flow data. Please include Ly-6G flow in part (C).
8. Statistical analysis; although this reviewer is not an explicit expert in statistics, it appears that SD is more appropriate than SEM in this paper. Please justify the use of SEM.
9. Suppl figure 4: How do the authors know that the anti-metastatic activity is due to killing of tumor cells (also comment3). Please address experimentally.
10. Suppl figure 4: Authors nicely provide evidence for the upper right arrow. However no direct evidence for the lower right arrow is provided (see comment 9). Only partial evidence for the upper left arrow is included. Authors suggest that primary tumors produce cytokines/chemokines to affect myeloid cells in the BM. However, they use injection of recombinant CCL3L1 to show this. It would be much more convincing to show experiments with tumors overexpressing ACKR2 ligands.
11. Indeed, the observation that ACKR2 deficiency results in increased primary tumor growth but reduced metastasis is surprising. Authors should discuss this and potential mechanisms (although speculative at this point) in more detail and provide potential explanations.

Minor comments:

1. Page 3 last paragraph; (Bonecchi and Graham, 2016) should be cited as upper case number.
2. Figure 2D/E: data are impossible to judge at this size and contrast of the figure. Could the authors perform IF instead? Please do not show Ly-6G area but rather absolute cell numbers (for example per optical field).

Reviewer #2:

(Remarks to the Author):

1. Some of your data are interesting. However, the links between your experimental approach are difficult to follow. The most interesting data are the growth of tumors in ACKR2^{-/-} mice (and metastasis) and the effect of the anti-Ly6G antibodies in the observed metastasis, that do suggest a role for granulocytes in metastasis. The rationale for the rest of the experimental approach is not well developed.
2. What do you envisage to be the mechanism through which the neutrophils may be influencing metastasis? You present data on the expression of various chemokine receptors, and then you study the expression of ACKR2 in hematopoietic precursors. Did you check the expression of ACKR2 in the

hematopoietic database of Stanford? Immgen indicates that it is also expressed in Pro-B cells. Any chance that these may influence your results?

3. Did you check the expression of cytokines that strongly influence neutrophils (like GCSF) in the ACKR2^{-/-} mouse?

Reviewer #3:

(Remarks to the Author):

In this study, the authors have investigated the role of Ackr2 on tumor behavior. Using Ackr2^{-/-} mice, it was shown that Ackr2 deficiency reduces the metastatic potential of experimental tumors, which was associated with increased numbers of monocytes and neutrophils in blood and lungs. In the B16 transplantation model, neutrophil depletion restored metastasis formation. Neutrophils from Ackr2^{-/-} exhibited an activated phenotype and increased expression of certain chemokine receptors. HPCs were shown to express Ackr2, which controlled expression of chemokine receptors. The authors conclude that ACKR2 in HPCs acts as a checkpoint of neutrophil release and anti-metastatic activity.

General points:

This study into the role of ACKR2 on the immune system and cancer behaviour has the potential to be very interesting, however, various concerns are raised.

1) The authors make various claims throughout the study that are not supported by the data presented here. Principally, the authors selectively switch between different cancer models and make the assumption that what they observe in one model would be recapitulated in other models. For example, the authors present the study as being in NeuT-driven primary mammary carcinogenesis, where Ackr2-deficiency resulted in neutrophil-mediated protection against metastasis (e.g. see abstract and discussion). This claim, however, is unsubstantiated as the only data demonstrating that neutrophils may have a role in metastasis protection is performed using B16-F10 cells (Fig. 4B) and not in the NeuT model. The authors should have included neutrophil depletion studies in the NeuT and 4T1 models, and they should have performed a thorough analysis of immune parameters in the B16 model in the WT and Ackr2^{-/-} setting. Another example of switching between tumor models without explanation is seen in Figure 5, where A-B are performed in B16 injected mice, and C in 4T1 injected mice.

2) This study offers very limited characterisation of the immune system of Ackr2-deficient mice (only monocytes/neutrophils are studied; the other immune cell populations are not analyzed) and

limited citations of previous studies that seek to characterise this mouse model, despite demonstrating that *Ackr2* expression is highest in LSK. This makes it very difficult to interpret data presented here. Furthermore, there is significant literature detailing haematopoietic perturbations in the *Ackr2*-deficient mouse model. In particular, in B1 cells, monocytes and neutrophils. Therefore, it is not possible to ascertain whether the observations presented here are a direct result of *Ackr2*-deficiency in neutrophils/neutrophil progenitors, or an indirect consequence of *Ackr2* being absent from all cells. The current study would be more convincing if *Ackr2* was conditionally deleted, to pinpoint its role either specifically in neutrophils or more broadly, in myelopoiesis.

3) This study does not solve the underlying mechanisms by which *Ackr2* deficiency changes neutrophils from being pro-metastatic into anti-metastatic. The authors show that neutrophils in 4T1-bearing *Ackr2*^{-/-} mice express higher ROS levels, and they speculate that this reveals a more active phenotype of neutrophils, however, the functional significance of this observation is not addressed. Furthermore, the authors demonstrate that granulopoiesis is increased in the *Ackr2*^{-/-} mice and assume that this leads to an accelerated maturation rate of neutrophils. However, they do not demonstrate that the neutrophils in *Ackr2*^{-/-} mice are mature.

4) Throughout the paper, the authors fail to give the vital controls of *Ackr2*^{-/-} mice compared to WT. For example, in Fig. 2C-F, it is important to show the circulating neutrophil and monocyte numbers in *Ackr2*^{-/-} mice compared to littermate controls without the NeuT background. Does lack of *Ackr2* itself drive increased release of these cells or does the added pressure on the system of NeuT tumors increase these cell numbers?

Specific points

In its current form, this paper is poorly written. The manuscript needs re-wording to be comprehensible, clear and precise. In addition, the introduction is not critically written. The authors cite one or two studies and assume that the results of these studies are fact.

Figure 1D, E and F: To make the claim that the impact of ACKR2 on tumour metastasis can be ascribed to its function in the host stromal compartment, it would be more informative to use a cell line in which *Ackr2* is knocked out, rather than one with low *Ackr2* expression prior to injection into the mouse. If the cell line still has functional *Ackr2*, it remains possible that *Ackr2* is upregulated by the tumor line once injected in vivo.

Figure 1A: this graph with double y-axes is confusing. In addition, it is not clear what the authors mean with tumor take (mass number). Probably this is the number of palpable or microscopic mammary tumors per mouse? Please explain.

Increased myeloid cell expansion in *Ackr2*^{-/-} tumor bearing mice: The authors state that there is myeloid expansion in these mice and cite previous work but they do not show it. The authors should

include the data demonstrating that myeloid expansion is actually happening in their system rather than just citing literature. Furthermore, with the 66cl4 cell line, it would be nice to see representative images of lung metastasis as in 1E.

Graphs 3E and 3F are mis-labeled on the y-axis: 3E should be neutrophils and 3F should be monocytes.

In many figures, particularly figures 4 and 5, it is not clear whether the data presented are from tumor bearing or non-tumor bearing mice.

Reviewer #1:

(Remarks to the Author):

Massara and co-authors investigate the role of an atypical chemokine receptor (ACKR2) in mediating metastasis of tumors. They propose a role for this receptor in modulating myeloid differentiation and subsequently the anti-metastatic activity of neutrophils. Authors use inactivation or downregulation of ACKR2 and injection in vivo models of recombinant ligands.

Suppl figure 4 summarizes the data. Whereas the paper and the proposed mechanism is of potential interest, more experimental data are needed to convincingly and sufficiently support the presented summary scheme and mechanism.

Specific major comments:

1. Page 6: Authors propose increased mobilization of myeloid cells in ACKR2-deficient mice. Does ACKR2-deficiency affect proliferation of myeloid cells and their precursors? It is important to show the respective experimental data on proliferation (and thus exclude this mechanism).

As the reviewer rightfully states, it is important to assess if ACKR2-deficiency is affecting hematopoietic progenitors' proliferation. We have performed in vivo experiments using a BrdU analog (EdU) to label proliferating cells and we have not found significant differences in the proliferation of hematopoietic progenitors LSK, CMP and GMP and MEP taken from WT and *Ackr2*^{-/-} mice. We have added this data in the result section and in the figure S6E.

2. On page 7 and in figure 4B authors report that depletion of myeloid cells reduces metastasis in WT mice but induces metastasis in ko mice. How is this explained? The authors do not make sufficient attempts to experimentally address this. They could for example compare the anti-metastatic potential of CD115+ cells and Ly-6G+ cells from WT and ko animals. How do they differ? It seems that this is a key question in this project?

As the referee correctly pointed out, in figure 4B and S4B we have reported that depletion of neutrophils and monocytes reduced metastasis in WT mice. When we did the same experiment in *Ackr2*^{-/-} mice, that were already protected from metastasis, we found that depletion of monocytes has no further protective effect, while ONLY neutrophil depletion rescued the phenotype of protection against metastasis in the B16F10 model indicating that neutrophils confer the protection in these mice. We have now shown in figure 4 only the results of neutrophil depletion adding the depletion we have performed with the 4T1 model that has given comparable results (Fig. 4C).

To further demonstrate that *Ackr2*^{-/-} neutrophils protect mice from metastasis, we have performed adoptive transfer experiments in which we directly compared the antimetastatic potential of WT and *Ackr2*^{-/-} Ly6G neutrophils (Fig. 4D). We found that the transfer of WT neutrophils did not modify metastasis number, while the transfer of *Ackr2*^{-/-} neutrophils protect the mice from metastasis (Fig. 4D). These cells were further phenotypically and functionally characterized (see next points).

3. Page 8, figure 5 and suppl Figure 4: Authors investigate ROS production and conclude that this is implicated in antimetastatic activity and tumor cell killing. To make this manuscript conclusive it is essential to perform tumor cell killing assays and to show that ROS is indeed involved in killing. Otherwise the experiments shown only present associations and not a mechanism as proposed in suppl figure 4. In addition, do neutrophils from WT and KO animals display different killing activity?

As suggested by the reviewer we have done in vitro killing assays with WT and *Ackr2*^{-/-} neutrophils. We have found that in an in vitro assay, WT neutrophils taken from tumor-bearing mice can kill about 20% of tumor cells and that this activity is inhibited by ROS inhibition. These results are consistent with previously published papers (Granot et al., Cancer cell, 2011).

When we did the same experiment with *Ackr2*^{-/-} neutrophils, we found that they have an increased killing activity that is inhibited by ROS inhibition. Similar results were obtained with neutrophils isolated from BM of WT and *Ackr2*^{-/-} mice and indicated that neutrophils taken from a KO mice have an increased killing activity. Results of these experiments are shown in figure 5K and S5D.

4. Page 9: Authors conclude that neutrophils are more efficiently recruited to metastatic lesions. Could the authors please address this experimentally for example by measuring the recruitment of adoptively transferred neutrophils from WT and KO. At present the authors do not really address the recruitment process.

As suggested by the reviewer, we have done adoptive transfer experiments using WT and *Ackr2*^{-/-} neutrophils and we have measured their recruitment to the lung in WT recipients using the CD45.1 and CD45.2 antigens to trace them. We have found that *Ackr2*^{-/-} neutrophils were more efficiently recruited to the lung with a concomitant decrease of transferred cells in the blood (results are now shown in figure 5H and 5I).

5. Figure 3: E/F show by BM transfer that expression of ACKR2 on neutrophils/BM affects cell number in blood. However, for this paper, the number of neutrophils in the lungs would be important (site of metastasis). It is important to repeat figure 3 E/F and measure neutrophils in the lungs.

As suggested by the reviewer, we have performed new BM chimera experiments to measure neutrophil recruitment in the lung. With these new experiments, we have confirmed results of increased myeloid cell mobilization in blood when BM cells were *Ackr2*^{-/-} (Fig. S3A and S3B) and observed an increased recruitment of neutrophils in the lungs only when BM cells were *Ackr2*^{-/-}. Results are now shown in figure S3C and S3D.

6. Figure 4: Please show absolute numbers of metastasis (not relative score) in order to be able to compare data to (A) and (C).

As the reviewer correctly pointed out in the submitted version of Figure 4, it was not possible to compare data for the different y axis used. In order to allow comparison of results shown in different figures of the article, we have now used the metastatic ratio (the ratio of metastasis in *Ackr2*^{-/-} or depleted mice compared to control mice). Indeed, even if the same batch of frozen cell was used, we have found variations in the absolute number of metastasis in the WT group but constant ratio of protection in *Ackr2*^{-/-} mice. This effect has been reported also in other papers (van der Weyden et al., Scientific data, 2017; van der Weyden et al., Nature, 2017). In any case, we provide the raw data of the single experiments performed in Tables S3, S4 and S5.

7. Suppl figure 1: Authors phenotype lung neutrophils without showing Ly-6G (key marker) flow data. Please include Ly-6G flow in part (C).

In supplementary figure 2 (in the previously submitted version Fig. S1), we have now included Ly6G labelling that demonstrate that the gated cells are neutrophils.

8. Statistical analysis; although this reviewer is not an explicit expert in statistics, it appears that SD is more appropriate than SEM in this paper. Please justify the use of SEM.

We thank the reviewer for raising this point to our attention. Indeed, SEM quantifies uncertainty in estimate of the mean while SD is the descriptive statistic to be used to indicate the dispersion of the data from mean. We now report our results showing SD instead of SEM.

9. Suppl figure 4: How do the authors know that the anti-metastatic activity is due to killing of tumor cells (also comment3). Please address experimentally.

As suggested by the reviewer, we have performed in vitro killing assays with WT and *Ackr2*^{-/-} neutrophils of tumor-bearing mice and resting BM neutrophils and found that *Ackr2*^{-/-} neutrophils have an increased killing activity compared to WT mice. These results are shown in figure 5K and S5D.

10. Suppl figure 4: Authors nicely provide evidence for the upper right arrow. However no direct evidence for the lower right arrow is provided (see comment 9). Only partial evidence for the upper left arrow is included. Authors suggest that primary tumors produce cytokines/chemokines to affect myeloid cells in the BM. However, they use injection of recombinant CCL3L1 to show this. It would be much more convincing to show experiments with tumors overexpressing ACKR2 ligands.

We have now provided evidence demonstrating the increased tumor killing activity of *Ackr2*^{-/-} neutrophils (comments 3 and 9) and we believe the lower right arrow is now adequately supported. Referring to the production of cytokines by the tumor (upper left arrow), high levels of colony-stimulating factors (G-CSF and GM-CSF), which promote granulopoiesis and neutrophil mobilization and are the main cause of the cancer-associated neutrophilia, have been reported in several tumor models, including the 4T1 model. We have now investigated this parameter in our experimental setting by measuring GM-CSF and G-CSF concentrations in the sera of WT and *Ackr2*^{-/-} mice. While GM-CSF was under the detection limit in both animals, we have found that both WT and *Ackr2*^{-/-} mice showed a strong increase in G-CSF levels after tumor implantation. These data are now reported in figure S6F.

We also provide a modified version of the figure (now numbered S7) based on the direct comparison between WT and *Ackr2*^{-/-} mice.

11. Indeed, the observation that ACKR2 deficiency results in increased primary tumor growth but reduced metastasis is surprising. Authors should discuss this and potential mechanisms (although speculative at this point) in more detail and provide potential explanations. We have now discussed this point in more detail in the discussion.

Minor comments:

1. Page 3 last paragraph; (Bonecchi and Graham, 2016) should be cited as upper case number.

We have corrected the citation.

2. Figure 2D/E: data are impossible to judge at this size and contrast of the figure. Could the authors perform IF instead? Please do not show Ly-6G area but rather absolute cell numbers (for example per optical field).

We have now increased the size of the figures and counted the absolute number of Ly6G on field of view (now shown in figure 2E).

Reviewer #2:

(Remarks to the Author):

1. Some of your data are interesting. However, the links between your experimental approach are difficult to follow. The most interesting data are the growth of tumors in ACKR2^{-/-} mice (and metastasis) and the effect of the anti-Ly6G antibodies in the observed metastasis, that do suggest a role for granulocytes in metastasis. The rationale for the rest of the experimental approach is not well developed.

We are now providing a revised manuscript in which we have substantially changed the text and better explained the rationale of our study. Briefly, in this paper we have analyzed the mechanism of metastasis protection in *Ackr2*^{-/-} mice. Having found increased number of myeloid cells in the blood and in the lungs of *Ackr2*^{-/-} mice and having found that this is due to an increased mobilization of these cells from the BM, we have performed selective antibody-dependent depletion experiments. We found that the protection is provided by Ly6G⁺ cells and we have formally demonstrated their ability to protect from metastasis performing adoptive transfer experiments. Furthermore, we have found that neutrophils which are present in the circulation of *Ackr2*^{-/-} mice show a more mature phenotype and increased effector functions. We have also found that these cells have increased expression of CC chemokine receptors, increased production of ROS and increased tumor killing activity. Finally, BM transfer experiments clearly pointed to a role of hematopoietic ACKR2 in the increased neutrophil mobilization and metastasis protection. Since neutrophils do not express ACKR2, we studied neutrophil BM progenitors and found that *Ackr2* is expressed by LSK cells and, to a minor extent, CMP and GMP, which showed increased expression of CCRs and increased myeloid differentiation. Finally, by expressing *Ackr2* in the HL60 cell line we found evidence that the impact of *Ackr2* on *Ccr2* and *Cd11b* expression operates in a cell autonomous fashion.

2. What do you envisage to be the mechanism through which the neutrophils may be influencing metastasis? You present data on the expression of various chemokine receptors, and then you study the expression of ACKR2 in hematopoietic precursors. Did you check the expression of ACKR2 in the hematopoietic database of Stanford? Immgen indicates that it is also expressed in Pro-B cells. Any chance that these may influence your results?

In the new version of the article we provide direct evidence that *Ackr2*^{-/-} neutrophils have increased tumor-killing activity (Fig. 5K and S5D) and show increased recruitment rate to the lung (Fig. 5H). The Immgen database reports low expression of *Ackr2* in hematopoietic progenitors and in pro-B cells, but B cell depletion experiments indicated that the protection from metastasis is B cell independent. Results are shown in figure S4C.

3. Did you check the expression of cytokines that strongly influence neutrophils (like G-CSF) in the ACKR2^{-/-} mouse?

As suggested by the reviewer we have measured the level of G-CSF and GM-CSF in the serum of WT and *Ackr2*^{-/-} mice both in basal and tumor conditions. GM-CSF was under the detection limit in both conditions, while the serum concentration of G-CSF, that is strongly increased in tumor bearing mice, is not different between WT and *Ackr2*^{-/-} mice (now shown in Fig. S6F).

Reviewer

(Remarks to the Author):

In this study, the authors have investigated the role of Acker2 on tumor behavior. Using Acker2^{-/-} mice, it was shown that Acker2 deficiency reduces the metastatic potential of experimental tumors, which was associated with increased numbers of monocytes and neutrophils in blood and lungs. In the B16 transplantation model, neutrophil depletion restored metastasis formation. Neutrophils from Acker2^{-/-} exhibited an activated phenotype and increased expression of certain chemokine receptors. HPCs were shown to express Acker2, which controlled expression of chemokine receptors. The authors conclude that ACKR2 in HPCs acts as a checkpoint of neutrophil release and anti-metastatic activity.

General points:

This study into the role of ACKR2 on the immune system and cancer behavior has the potential to be very interesting, however, various concerns are raised.

1) The authors make various claims throughout the study that are not supported by the data presented here. Principally, the authors selectively switch between different cancer models and make the assumption that what they observe in one model would be recapitulated in other models. For example, the authors present the study as being in NeuT-driven primary mammary carcinogenesis, where Acker2-deficiency resulted in neutrophil-mediated protection against metastasis (e.g. see abstract and discussion). This claim, however, is unsubstantiated as the only data demonstrating that neutrophils may have a role in metastasis protection is performed using B16-F10 cells (Fig. 4B) and not in the NeuT model. The authors should have included neutrophil depletion studies in the NeuT and 4T1 models, and they should have performed a thorough analysis of immune parameters in the B16 model in the WT and Acker2^{-/-} setting. Another example of switching between tumor models without explanation is seen in Figure 5, where A-B are performed in B16 injected mice, and C in 4T1 injected mice.

We thank the reviewer for this comment and put effort in the amended version to improve the manuscript with respect to this critical issue. We now show a complete analysis of immune parameters in the B16F10 model in Fig. 5 A-E (and S4A) and moved the analysis of the same parameters in the 4T1 model in Fig. S5A-C. In order to provide and substantiate our results we have also performed several additional in vivo experiments. We have confirmed the role of neutrophils in the protection from spontaneous lung metastasis in Acker2^{-/-} mice using the 4T1 orthotopic tumor model by performing also in this model depletion experiments (results in figure 4C). We have also amended the manuscript (abstract and discussion) to clearly state in which models we have seen the neutrophils-mediated protection.

2) This study offers very limited characterisation of the immune system of Acker2-deficient mice (only monocytes/neutrophils are studied; the other immune cell populations are not analyzed) and limited citations of previous studies that seek to characterise this mouse model, despite demonstrating that Acker2 expression is highest in LSK. This makes it very difficult to interpret data presented here. Furthermore, there is significant literature detailing haematopoietic perturbations in the Acker2-deficient mouse model. In particular, in B1 cells, monocytes and neutrophils. Therefore, it is not possible to ascertain whether the observations presented here are a direct result of Acker2-deficiency in neutrophils/neutrophil progenitors, or an indirect consequence of Acker2 being absent from all cells. The current study would be more convincing if Acker2 was conditionally deleted, to pinpoint its role either specifically in neutrophils or more broadly, in myelopoiesis.

We apologize with the reviewer for not having provided references to some key papers which have already reported an extensive immune parameters characterization in Acker2^{-/-} mice. These

references, including our own paper on Blood 2012, have now been added (Savino et al., Blood 2012; Hansell et al., Blood, 2011).

As the reviewer correctly pointed out, these papers indicate that *Ackr2*^{-/-} mice have perturbations in the number and in the function of monocytes and B cells. For this reason, we have performed depletion experiments of these two leukocyte subpopulations (now shown in Fig. S4B and S4C), whose results rule out their role in the metastasis protection we are here reporting.

We agree with the referee that a conditional deleted mouse model would be of help, but at the moment ACKR2 floxed mice are not available. Furthermore, as our results indicate that the impact on neutrophils is caused by the lack of activity of *Ackr2* in hematopoietic progenitors, its deletion in this cellular compartment would not fully clarify the issue. To overcome this problem, we have performed BM chimeric mice (Fig. S1C) and adoptive transfer experiments (Fig. 4D) that demonstrate that neutrophils are the protective cell population.

3) This study does not solve the underlying mechanisms by which Ackr2 deficiency changes neutrophils from being pro-metastatic into anti-metastatic. The authors show that neutrophils in 4T1-bearing Ackr2^{-/-} mice express higher ROS levels, and they speculate that this reveals a more active phenotype of neutrophils, however, the functional significance of this observation is not addressed. Furthermore, the authors demonstrate that granulopoiesis is increased in the Ackr2^{-/-} mice and assume that this leads to an accelerated maturation rate of neutrophils. However, they do not demonstrate that the neutrophils in Ackr2^{-/-} mice are mature.

In order to give functional relevance to our observations, we have performed in vitro tumor killing experiments demonstrating that *Ackr2*^{-/-} neutrophils have increased killing activity compared to WT neutrophils (Fig. 5K and S5D). Furthermore, we demonstrated that adoptive transfer of these cells in tumor bearing mice inhibit lung metastasis (Fig. 4D).

Referring to granulopoiesis, we did not find increased proliferation of hematopoietic and myeloid progenitors in the BM of *Ackr2*^{-/-} mice (Fig. S6E). However, we have found a different phenotype of circulating neutrophils in *Ackr2*^{-/-} mice (higher levels of ICAM-1 and inflammatory CC chemokine receptors levels, lower CD62L levels, higher ROS production; Fig. 5A to 5G) that are markers of neutrophil activated or aged phenotype. In order to formally demonstrate that, despite similar granulopoiesis rate, *Ackr2*^{-/-} neutrophils are more mature, we have performed additional in vivo experiments following protocols using BrdU (Casanova-Acebes et al., Cell, 2013). We found an increased percentage of mature neutrophils in the blood of *Ackr2*^{-/-} mice compared to WT mice. We have added the results of this experiments in figure S6I.

4) Throughout the paper, the authors fail to give the vital controls of Ackr2^{-/-} mice compared to WT. For example, in Fig. 2C-F, it is important to show the circulating neutrophil and monocyte numbers in Ackr2^{-/-} mice compared to littermate controls without the NeuT background. Does lack of Ackr2 itself drive increased release of these cells or does the added pressure on the system of NeuT tumors increase these cell numbers?

We again apologize with reviewer not having shown the vital controls. As suggested, we have added them in the number of circulating and infiltrating leukocytes in the NeuT model (Fig. 2A and B and S2C) and in figure 5B and D and S5A and B to demonstrate differences found in neutrophils are due to the combined effect of the tumoral condition and ACKR2-deficiency.

Specific points

In its current form, this paper is poorly written. The manuscript needs re-wording to be comprehensible, clear and precise. In addition, the introduction is not critically written. The authors cite one or two studies and assume that the results of these studies are fact.

We have rewritten the paper (new parts are in red) to be more comprehensible and precise. We have rewritten the introduction and added more citations.

*Figure 1D, E and F: To make the claim that the impact of ACKR2 on tumour metastasis can be ascribed to its function in the host stromal compartment, it would be more informative to use a cell line in which *Ackr2* is knocked out, rather than one with low *Ackr2* expression prior to injection into the mouse. If the cell line still has functional *Ackr2*, it remains possible that *Ackr2* is upregulated by the tumor line once injected in vivo.*

In order to investigate the hypothesis that ACKR2 is induced in tumor cells once injected, we have sorted CD45⁺/CD31⁻/Podoplanin⁻ cells from disaggregated 4T1 tumors and performed qPCR for ACKR2 expression. We found that the ACKR2 is not upregulated on tumor cells in vivo. Results are now shown in figure S1B.

Figure 1A: this graph with double y-axes is confusing. In addition, it is not clear what the authors mean with tumor take (mass number). Probably this is the number of palpable or microscopic mammary tumors per mouse? Please explain.

We have now done two different graphs to avoid the confusing effect of the double y axis. We have maintained in the figure 1A the tumor volume and moved in figure S1A the tumor take (that is the number of mammary tumors per mouse, now explained in the material and methods section).

*Increased myeloid cell expansion in *Ackr2*^{-/-} tumor bearing mice: The authors state that there is myeloid expansion in these mice and cite previous work but they do not show it. The authors should include the data demonstrating that myeloid expansion is actually happening in their system rather than just citing literature. Furthermore, with the 66cl4 cell line, it would be nice to see representative images of lung metastasis as in 1E.*

We apologize for not giving the right controls of our mice in order to demonstrate the increased number of myeloid cells in the tumor-bearing *Ackr2*^{-/-} mice. Now we have added all the controls and corrected the text and shown data on the increased number of myeloid cells with all the controls in figure 2A and 2B. Referring to the 66cl4 cell line, we have decided to not show representative lung metastasis images because at the magnification of Fig 1E they are not visible.

Graphs 3E and 3F are mis-labeled on the y-axis: 3E should be neutrophils and 3F should be monocytes.

We apologize for the mistake, we have corrected the y-axis in the figures 3E and 3F.

In many figures, particularly figures 4 and 5, it is not clear whether the data presented are from tumor bearing or non-tumor bearing mice.

In all the figures, we have better specified which data derive from tumor or non-tumor bearing mice.

Reviewer #1:

Remarks to the Author:

I thank the authors for precisely and specifically addressing most of the issues raised during the review by my colleagues and by myself. The manuscript is now greatly improved and conclusive.

Only one minor criticism remains:

I disagree that the data provide convincing evidence for an activated neutrophil phenotype in the KO mice.

Fig 5. CD62L and ICAM expression is only marginally induced and does not convincingly demonstrate an activated neutrophil phenotype. I suggest removing these data from the manuscript and modifying the text accordingly. I also suppose that expression levels of CD62L (fragile flow marker especially after tissue dissociation and dissection!) and ICAM are not essential for the major message of this very nice paper.

Sven Brandau

Reviewer #2:

None

Reviewer #3:

Remarks to the Author:

In the revised manuscript, the authors have sufficiently addressed the raised concerns. They have added new experimental data that significantly strengthen the conclusions of the manuscript. In addition, the authors have considerably improved the writing style.

POINT-BY-POINT RESPONSE TO REFEREES

REVIEWERS' COMMENTS:

Reviewer #1 (Remarks to the Author):

I thank the authors for precisely and specifically addressing most of the issues raised during the review by my colleagues and by myself. The manuscript is now greatly improved and conclusive.

Only one minor criticism remains:

I disagree that the data provide convincing evidence for an activated neutrophil phenotype in the KO mice. Fig 5. CD62L and ICAM expression is only marginally induced and does not convincingly demonstrate an activated neutrophil phenotype. I suggest removing these data from the manuscript and modifying the text accordingly. I also suppose that expression levels of CD62L (fragile flow marker especially after tissue dissociation and dissection!) and ICAM are not essential for the major message of this very nice paper.

Sven Brandau

Following the suggestion of the reviewer we have removed data from Figure 5 and supplementary figure 5 on the neutrophil phenotype and modified the text.